

# Automatic processing of atmospheric CO$_2$ and CH$_4$ mole fractions at the ICOS Atmospheric Thematic Center

Lynn Hazan, Jérôme Tarniewicz, Michel Ramonet, Olivier Laurent, Amara Abbaris

Laboratoire des Sciences du Climat et de l'Environnement (LSCE/IPSL), UMR CEA-CNRS-UVSQ, Gif-sur-
Yvette, France

*Correspondence to*: L.Hazan (lynn.hazan@lsce.ipsl.fr)

**Abstract.** The Integrated Carbon Observation System Atmospheric Thematic Center (ICOS ATC) automatically
processes atmospheric greenhouse gases mole fractions of data coming from sites of the ICOS network. Daily
transferred raw data files are automatically processed and archived. Data are stored in the ICOS atmospheric
database, the backbone of the system, which has been developed with an emphasis on the traceability of the data
processing. Many data products, updated daily, explore the data through different angles to support the quality
control of the dataset performed by the principal operators in charge of the instruments. The automatic
processing includes calibration and water vapour corrections as described in the paper. The mole fractions
calculated in near-real time (NRT) are automatically revaluated as soon as a new instrument calibration is
processed or when the station supervisors perform quality control. By analyzing data from eleven sites, we
determined that the average calibration corrections are equal to 1.7 ±0.3 µmol mol$^{-1}$ for CO$_2$ and 2.8 ±3
nmol mol$^{-1}$ for CH$_4$. These biases are important to correct to avoid artificial gradients between stations that could
lead to error in flux estimates when using atmospheric inversion techniques. We also calculated that the average
drift between two successive calibrations amounts to ±0.05 µmol mol$^{-1}$ and ±0.7 nmol mol$^{-1}$ for CO$_2$ and CH$_4$,
respectively. Outliers are generally due to errors in the instrument configuration and can be quickly detected
thanks to the data products provided by the ATC. Several developments are still ongoing to improve the
processing, including automated spike detection and calculation of time-varying uncertainties.

**1    Introduction**

Rising greenhouse gas (GHG) concentration in the atmosphere is a major source of forcing in the current
changing climate (Intergovernmental Panel on Climate Change, 2013). Worldwide measurement systems are
being implemented (Andrews et al., 2014; Deng et al., 2014; Deutscher et al., 2014; Dils et al., 2014; Fang et al.,
2014; Frankenberg et al., 2014; Houweling et al., 2014; Ramonet et al., 2010) to both monitor and understand
these increasing concentrations. In Europe, the Integrated Carbon Observation System (ICOS), an international
research infrastructure for precise in situ measurements, has been built. ICOS is a distributed infrastructure
composed of three integrated networks measuring GHG in the atmosphere, over the ocean and at the ecosystem





level. Each network is coordinated by a thematic center that performs, among other things, centralized data processing. Further processing takes place in the ICOS Carbon Portal where, for example, 2D GHG flux maps are computed using the ICOS atmospheric station time series. One of the key focuses of ICOS is to provide standardized and automated high-precision measurements, which is achieved through the use of measurement

protocols and standardized instrumentation. The implementation of ICOS included a preparatory phase (2008–2013, EU FP7 project reference 211574) with a demonstration experiment, later called "extended demo experiment" in the period between the end of the preparatory phase and the formal start of ICOS as a legal entity at the end of 2015. In total, eleven sites have been participating in the atmospheric network during this demonstration experiment and its extension. The data center of the ICOS ATC, located at the Laboratoire des

Sciences du Climat et de L'Environnement (LSCE, France), began to automatically process atmospheric greenhouse gas mole fractions in 2009. The centralized data processing aims to reduce inter-laboratory differences and facilitate the production of a coherent dataset in near-real time; NRT is defined here as on a daily basis.

NRT data production is more demanding but brings several benefits. In terms of station management, it allows

station principal operators and investigators to get a fast feedback on the data; it improves reactivity in case of disruption in the data flow and thus limits data gaps. NRT data are also useful for campaign-based measurement setups. It allows, for example, adjustment of the campaign setup, to correct for the observation plan or place more emphasis on a specific phenomenon. On a more scientific level, NRT data allow for early warning monitoring systems, for example, in the case of extreme GHG events (e.g., drought, high-pollution event). NRT

is a necessity to perform data assimilation for operational systems (e.g., Monitoring Atmospheric Composition & Climate—MACC) where NRT data are either used as a diagnostic or ingested in assimilation mode to improve operational forecasting (http://www.gmes-atmosphere.eu/d/services/gac/verif/ghg/icos).

NRT data are, however, less precise than so-called consolidated data. In ICOS, consolidated data are expected to be produced on a 6-month basis. They contain additional data treatment steps ensuring increased precision and

confidence in the dataset. These steps include potential correction due to drift in the reference scales used to make the measurements and "manual visual" inspection of the data to screen for potential problems that are difficult to detect automatically.

To further increase confidence and trust, ICOS is building an efficient scheme to ensure traceability of the data. Persistent identifiers (PID) will be attached to the data for both proper acknowledgment and citation. ICOS

atmospheric data are traceable to the Global Atmosphere Watch (WMO/GAW) international reference scales for GHG, and the history of data processing steps is archived. This allows full traceability and transparency of the consolidated dataset, which will be the basis for elaborated products and services.

This article describes the computing facility dedicated to the ICOS ATC at LSCE, the different steps of the automatic processing of $CO_2$ and $CH_4$ mole fractions including the automatic quality control of the raw data, and

the corrections due to water vapour interference and calibration (WMO scale). Most of the processing protocols and parameters are illustrated with a few examples from instruments currently providing raw data to the ICOS





ATC as part of the ICOS extended demonstration experiment. Because the paper is focused only on $CO_2$ and $CH_4$, the analyzers that are currently deployed in the monitoring network have been considered. To date, for these species, only cavity ring down spectroscopy (CRDS) analyzers commercialized by the Picarro company meet the ICOS requirements, but other instruments may be added in the future.

## 2    Server organization and data archive at ICOS ATC

The instrumental raw data are transferred at least once a day from the monitoring sites to an ATC server using the Secure File Transfer Protocol (SFTP). The files are first archived, and the data are automatically processed by the ICOS database. Three dedicated servers (Figure 1) are installed and maintained at ICOS ATC to fulfil automatic data collection from measurement stations, processing and distribution to users.

- Data collection (icos-ssh server)—ICOS network stations daily upload raw data from the instruments to the ATC. This upload can be managed by upload software developed at ATC. All collected data are centralized on the icos-ssh server and upon receipt are copied to a dedicated server for archival. Data are kept on the icos-ssh server for one month after their upload. A duplicate archive of the raw data is under study with the ICOS Carbon Portal. Currently, the amount of data uploaded is on the order of 6.5 MB/day

15     per $CO_2/CH_4$ in situ analyzer, corresponding to a total ~170 MB a day for 26 stations processed daily by ATC. Note that the files transferred every day to the ATC are not the high-resolution absorption spectra used to retrieve mole fractions (Crosson, 2008). The raw data files of the trace gas analyzers currently processed at ATC contain $CO_2/CH_4$ information already in geophysical units. It is foreseen that the full spectra files will be archived at the ICOS site on specific hard drives for further post-analysis. The

amount of data to archive would then be approximately 230 MB/day, 780 MB/day and 1.3 GB/day, respectively, for models ESP100/G1301, G2301 and G2401 of CRDS Picarro analyzers.

- Data processing (icos-data server)—Upon reception, data are processed. The processing is performed on the icos-data server, a dedicated internal (inaccessible from outside the ATC) server at ATC that also hosts the ICOS atmospheric database. The icos-ssh server (accessible from outside the ATC) also hosts the

QA/QC applications developed at ATC, used by principal investigators (PIs) and authorized persons to carry out the measurement control.

- Data distribution (icos-web server)—The distribution of data and data products are served by the icos-web server. This server hosts the ATC website and uses the Drupal open-source content management system framework. For security purposes, only read-only access is allowed to some partitions on the icos-

data server. Access to the ICOS atmospheric database hosted on icos-data from icos-web is prohibited.

Traceability of the data downloads and long-term archival, which are not described here, are being implemented in collaboration with the Carbon Portal of ICOS, which is hosted and operated at Lund University in Sweden (https://www.icos-cp.eu/).



### 3    Processing: automatic filtering of raw data

Specific processing chains are developed for each type of trace gas analyzer, but the general framework remains the same. Here, we describe the processing chain and associated parameters defined for the treatment of continuous measurements of $CO_2$ and $CH_4$ atmospheric mole fractions. Similar chains are developed for

measurements of other ICOS parameters such as meteorological variables or radon but are not described in detail in this article. Figure 2 gives an overview of the different steps of the $CO_2$ and $CH_4$ data processing. One analyzer routinely measures three types of air samples: ambient air, air from target tanks and air from calibration tanks. The target tanks, also called 'surveillance tanks', are used as a quality control tool. Their mole fractions are known (prescribed by the ICOS Central Analytical Laboratories (CAL) located in Germany, which is in charge

of providing the calibration gases needed by the atmospheric stations) and are processed similarly to the ambient air. Consequently, the temporal variations of the target gas measurements can be used to estimate time-varying uncertainties (Yver-Kwok et al., 2015). It should be noted, however, that the target gases do not pass through the whole air inlet, and possible bias due to a contamination in the inlet upstream the connection of the target gas is not considered. As recommended by WMO, two target tanks, with a significant range in the mole fractions of the

measured species, are required at ICOS stations (WMO, 2012). Short-term target gases are analyzed at least once per day, whereas long-term target gases are measured only once every 2 to 4 weeks (after each calibration sequence). This configuration allows for both frequent measurements using one target gas and the possibility to keep the other target gas over a long period (10 to 20 years). The system also handles so-called inter-comparison (ICP) tanks, which correspond to cylinders analyzed as part of a comparison exercise like the round-robin set up

by WMO/GAW or by the Integrated non-$CO_2$ Greenhouse Gas Observing System (InGOS) European project (Manning et al., 2009; WMO, 2012). The ICP gases are processed similarly to target gases. The processing of the different types of gas follows the same general scheme: data control, correction, filtering and time aggregation (Figure 2).

For traceability and transparency of the data processing, each rejection of data is associated with a flag. For this

purpose, an internal cumulative flag has been defined, which is associated with the different steps of the processing. The steps and the flag will be described in the following paragraphs. Because each instrument may have different ways and conditions to validate their raw data, the internal flags are instrument dependent. If these flags are important for the traceability of the process, they are inconvenient for the majority of the data users who request a simple and unambiguous way to separate the valid and invalid data. For this reason, we have

defined another flag scheme named 'user flag', as described in Table 1. It is instrument independent and allows easy differentiation of the data that have been validated/invalidated either through NRT data processing or after the requested inspection of the data by an expert. This flagging scheme is completed with a third type of flag named 'descriptive flag', which allows the PI to provide codified reasons for invalidating data or useful information for validating data. For each data point, there is an automatic descriptive flag and a manual

descriptive flag. The manual flag is set by the expert via a graphical quality control application, and the automatic flag is set during the automatic processing of the data. Both flags use the same list of possible values.



The flags are set only on raw data. The flag information on raw data is carried to the aggregated data (minute or hourly averaged or injection). A description of the 'descriptive flag' can be found in Table 2.

### 3.1 System configuration

The objective of ICOS is to develop a standardized European monitoring network for greenhouse gases with centralized data processing. Technical discussions about the measurement protocols have been organized during the ICOS preparatory phase through seven working groups. This process has resulted in the first version of the ICOS Atmospheric Station Specifications (ICOS-MSA, 2014). Because the monitoring stations have specific local constraints, it has been required that the processing chains can be parameterized to handle some of the station specificities. A dedicated application, called ATCConfig, has been developed to allow the station PIs to configure the stations of which they are in charge. This application enables the following key points to be described in detail:

- Contact persons and institutes in charge of the station and instruments

- Geographic coordinates, postal address and description of the monitoring station including the different measurement setups with plumbing schemes

- Instrument description: category, model, firmware, location to trace instrument movements (e.g., for reparation), various related metadata

- Calibration/target tanks: model, date of inspection, valve and regulator description, filling date and mole fractions values

- Description of the sampling line connections and tank connections to the instruments

- Description of the measurement sequences (in situ air, calibration and target gases; see Table 3)

- Definition of the measurement processing parameters (control, correction and data filtering; see Table 4)

Each registered instrument is assigned a unique identifier used to reference it (preceded with the # sign in this article). A key aspect of the designed system is to ensure a high level of traceability that leads us to keep track of the history of all configurations provided by station PIs.

Regarding the configuration of the measurement processing, we consider three types of sequences: calibration, ambient/target and inter-comparison. Table 3 provides the list of parameters for each of the three sequences, with the Mace Head station (identified by the 3-letter code MHD) configuration as an example. The station PIs must configure what is measured (tanks or in situ air), in which order and for how long. Minimum requirements—e.g., at least three calibration tanks and two target gases—are prescribed by an ICOS Atmospheric Station Specifications document.

The full list of parameters to be set up by station PIs for the operation of in situ $CO_2/CH_4$ analyzers is shown in Table 4, with the example of the Mace Head set of values for instrument #41. The means by which those





parameters are used in the automatic processing of the raw measurements of $CO_2$ and $CH_4$ mole fractions is described in the following paragraphs.

### 3.2    Control based on analyzer ancillary data

The first step of the processing consists of the evaluation of instrumental parameters (e.g., temperature, pressure,

flowrate). In the case of the $CO_2/CH_4$ analyzers currently used in the ICOS network, we are scanning each raw data point for three parameters: the cavity pressure, the cavity temperature and the outlet valve opening. These ancillary data are provided by the analyzer at the same time resolution as the raw $CO_2$ and $CH_4$ data. Consequently, for each single data point, we verify that the values of the parameters are in agreement with the prescribed physical parameters. An example of the range of variability allowed for those parameters, for

instrument #41 at Mace Head, is provided in Table 4. The valid intervals and thresholds are instrument and location dependent at this point, but discussions are ongoing between the scientists in charge of the instruments to evaluate the possibility to standardize these criteria for a given instrument model. This decision depends on whether the setup of the station has an influence on the instrument performance. For each GHG data measurement, all selected parameters are tested against their valid interval or threshold. If at least one parameter

fails, the GHG data are flagged as invalid. Each failure is traced in the internal cumulative flag (Table 5).

Table 6 shows all internal flags that have been attributed to three analyzers continuously measuring the $CH_4$ mole fractions during 2014. From this list, it appears that raw data may be rejected for a combination of reasons. For example, during the stabilization period following the switch from one gas to another, the cavity pressure and temperature may also be out of the assigned validity range. Overall, for an instrument working without

major failure, as in the case for the instruments in Table 6, the major cause of data rejection corresponds to the flushing time needed to stabilize the measurement after a change in the type of gas to analyze (e.g., from ambient air to target gas). Typically for a surface site with a single sampling level, the amount of data rejected for stabilization is on the order of 1 to 2% of the continuous raw data. For a multiple sampling level site, such as the Observatoire Pérenne de l'Environnement (identified by the 3-letter code OPE) high tower in France, this

percentage of rejected data can increase to 16% (Table 6) because of the frequent changes from one sampling level to another.

### 3.3    Control of the stabilization periods

When the instrument switches between sample types or sampling levels, some residual gas remains in the common tubing and valves. For a given duration (called the stabilization period) after such switches, the data are

flagged as invalid to avoid considering residual or mixed gas for further processing. The stabilization period duration depends on the flowrate, the volume of the analyzer cell, and the volume of the sampling line where continuous flushing is impossible. Consequently, the duration of the stabilization, given in minutes, is instrument and site dependent. Different values for the flushing time can also be set for in situ measurements and tank (calibration and target gas) measurements.



An example of the stabilization of $CO_2$ and $CH_4$ mole fractions is provided in Figure 3, showing a synthesis of the calibration gas measurements at Amsterdam Island station (identified by the 3-letter code AMS, whose instrument identifier is 111). At this station, four calibration gases are analyzed every 30 days four times for 30 min. The $CO_2$ and $CH_4$ mole fractions are averaged every minute, and we calculate the differences with the last

minute of each target injection. On average, stabilization ($\pm 0.05$ µmol mol$^{-1}$ for $CO_2$ and $\pm 1$ nmol mol$^{-1}$ for $CH_4$) is reached after 2 to 4 min. When looking at measurements of short-term and long-term target gases from several sites (Figure 4), one can see that stabilization is very often reached within 10 min, but more time may be needed for the long-term target. The difference can be explained by the fact that the long-term target is used only once a month, and the associated pressure regulator and lines must be flushed for a little while before being stabilized.

**4    Corrections of the $CO_2$ and $CH_4$ mole fractions**

The second step of the processing consists of correcting the data (Figure 2) for several artifacts. Corrections are applied only to the raw data that have been flagged as valid during the first step (see section 3). This also implies, in case of multiple corrections, that data flagged as invalid after one correction will not have the second correction function applied to them. This step is common to all types of gas, but the list of corrections to apply

differs among gas types (ambient air, target or calibration gases). There can be zero to n correction(s) where the order in which they are applied is meaningful. For each type of correction, there is a correction function defined, and the parameterization of this function is instrument, location, species and type of gas dependent.

For $CO_2$ and $CH_4$ measurements, all types of samples (ambient air, target, and calibration) are corrected to consider the humidity, and the calibration gases are not corrected by the calibration equation.

**4.1    Water vapour correction**

To achieve the WMO/GAW compatibility goals for observations of $CO_2$ and $CH_4$ mole fractions in dry air, it is required when using gas chromatography or non-dispersive infrared spectroscopy, to dry the air sample prior to analysis to a dew point of no more than -50°C (WMO, 2012). The emergence of new instruments using infrared absorption at specific spectral lines selected to minimize the interference between $CO_2$/$CH_4$ and water vapour

has enabled precise measurements in humid air. This technology, including cavity ringdown spectroscopy (CRDS) or cavity enhanced absorption spectroscopy, has been evaluated in both laboratory and field conditions by several research groups (Chen et al., 2010; Rella et al., 2012). Those studies have demonstrated that it is possible to precisely correct the effects of water vapour dilution and pressure broadening for $CO_2$ and $CH_4$. An empirical quadratic correction has been established by Chen et al. (2010) for CRDS Picarro analyzers and

confirmed by other laboratory experiments. In our data processing, the water vapour correction is applied in the same way to all analyzed samples (calibration and target gases, ambient air). By default, we are using the parameters defined by Chen et al. (2010):

$$CO_{2\,dry} = \frac{CO_{2\,wet}}{1 - 0.012 \times H - 2.674 \times 10^{-4} \times H^2}$$

(1)





$$CH_{4dry} = \frac{CH_{4wet}}{1 - 0.00982 \times H - 2.393 \times 10^{-4} \times H^2}$$

(2)

where $CO_{2wet}$ and $CH_{4wet}$ are the mole fractions measured in wet air, H the reported $H_2O$ mole fraction, and $CO_{2dry}$ and $CH_{4dry}$ the mole fractions in dry air.

The ICOS processing system allows the parameters of the quadratic equation to be changed to improve the water

vapour correction for each specific instrument. The setup of specific parameters for one instrument requires laboratory experiments to be performed as described by Rella et al. (2012). Such experiments are now performed systematically for each ICOS instrument at the ATC ICOS Metrology Laboratory. A technical paper describing these tests and associated results is in preparation.

Figure 5 shows a comparison of the water corrections applied to $CO_2$ and $CH_4$ measurements on two instruments

running in parallel at the Mace Head station. One instrument (G1301 model, #41) is directly measuring the wet air, whereas for the other one (G2301 model, #54), the air is preliminary dried with a cryogenic dryer using a 'cold trap' immersed in an ethanol bath cooled at -50°C. The $H_2O$ measurements decrease from approximately 1% (wet air) to less than 0.01% (dry air). The mean water vapour corrections applied in February 2014 for the instrument measuring the ambient air without any drying are 4.6±0.7 µmol mol[-1] and 17.8±2.8 nmol mol[-1],

respectively, for $CO_2$ and $CH_4$ (Figure 6). The same corrections applied to the instrument measuring dry air are 0.04±0.01 µmol mol[-1] and 0.16±0.05 nmol mol[-1], respectively, for $CO_2$ and $CH_4$. Overall, over the 15-day period shown in Figure 6, the differences between the dry mole fractions measured by the two instruments (#41 minus #54) at the Mace Head station are +0.015 ±0.03 µmol mol[-1] and -0.41 ±0.3 nmol mol[-1], respectively for $CO_2$ and $CH_4$.

We have made the same calculations for the differences between the $CO_2$ and $CH_4$ mole fractions before and after the water correction for eleven instruments used at monitoring stations in 2014. Statistics of the comparisons of hourly means over the year are summarized in Figure 7. Several instruments are operated with a drier system, and the water vapour corrections are consequently close to zero, as shown for the Mace Head station. For the other instruments, the water vapour corrections range for annual averages from 4 to 12 µmol mol[-

1] for $CO_2$ and from 18 to 40 nmol mol[-1] for $CH_4$, depending on the mean water vapour content. For example, the lowest corrections are observed at Pic du Midi station (identified by the 3-letter code PDM ), which is a high-altitude station (2877 m) with drier air compared to low-elevation stations. The statistics of the Trainou station (identified by the 3-letter code TRN) instrument #108 are intermediate between the dry and wet instruments because this instrument was operated in both situations in 2014.

**4.2    Calibration correction**

All $CO_2$ and $CH_4$ measurements that are intended to be added to the international monitoring networks database must be calibrated relatively to the WMO mole fraction scale for gas mole fractions in dry air maintained by WMO/GAW Central Calibration Laboratories (CCL). The current scales for $CO_2$ and $CH_4$ are 'WMO $CO_2$



X2007' (http://www.esrl.noaa.gov/gmd/ccl/co2_scale.html) and 'WMO CH$_4$ X2004' (http://www.esrl.noaa.gov/gmd/ccl/ch4_scale.html). As explained previously (see 3.1), the station PIs are in charge of the configuration of the calibrations performed at their site (number of calibration tanks, frequency of calibrations, and duration of the gas injections).

A calibration episode is called a "calibration sequence". When n working standards (calibration tanks) are measured in a row, the succession of tanks in a defined order is called a cycle. During a calibration episode, the cycle is repeated several times, and the calibration sequence is defined as m times the repetition of the unitary cycle element (Figure 8).

For each tank and each cycle, 1min mole fraction means are calculated, and the injection mean is derived from
the average of all minute means over the entire sampling period (exception made of the stabilization period). For each tank, the mole fraction means are then averaged over all m cycles. These values are plotted against the tank's standard concentration attributed by the calibration laboratory, and the calibration equation is determined by curve fitting using linear least square functions.

Because the calibration correction is essential for the final in situ or target data value determination, the
calibration data are filtered through a set of specific controls to determine whether all expected data are present and the quality is sufficient for use in the computation of the calibration equation (see below). All controls made on the calibration sequences are instrument, location and species dependent. If there are enough valid data, the calibration is accepted, and the calibration equation is determined. The equation coefficients are stored in the database, making them available for the calibration of the other types of samples (ambient air and target gases).

The controls applied to the calibration data are currently the following:

1. The expected number of cycles with their associated number of calibration standards is checked along with the minimum duration of the tank injection. If the calibration data do not correspond to the defined calibration sequence, the calibration is not taken into account.

2. The standard deviation of mole fraction minute means must be below a specified threshold.

3. The standard deviation of mole fraction injection means must be below a specified threshold.

4. A stabilization period given in terms of numbers of cycles can be applied.

5. The number of valid calibration injections (or cycle means) for each working standard, after applying the cycle stabilization, if any, must be equal to or greater than a minimum.

6. The number of valid working standard mole fraction means for the entire calibration sequence to use for
the computation of the calibration equation must be equal to or greater than a minimum.

An example of calibration for instrument #41 at Mace Head station on December 10, 2014, is shown in Figures 9 and 10. The set of parameters defined by the PI for this instrument are given in Tables 3 and 4. Four calibration tanks are used and are analyzed four times (cycles) for 20 min in each calibration, including 15 min dedicated to





the flush of the inlet lines and analyzer cell (stabilization time). Overall, the calibration lasts for 320 min. Figure 8 shows the different steps of the calibration process from analyzing the raw data and aggregating to the minute to calculating the cycle and calibration sequence averages. A fitting function (see fig 9) is then applied to the results of the calibration to define the coefficients of the correction, which will be applied to in situ air and target

gas measurements to ensure the data are compatible with the WMO reference scales.

Similar to the analysis of the water vapour corrections, we have summarized the calibration corrections applied at eleven instruments in 2014 (Figure 10). All stations are calibrated with standard gases, which are themselves measured against the international WMO scales. The correction applied to the raw data depends on the pre-set calibration parameters of the CRDS analyzers, which correspond in this study to the factory settings. The mean

$CO_2$ correction applied to the eleven instruments is 1.7 ±0.3 µmol mol$^{-1}$, and its variability over a one-year period, expressed as the mean standard deviation, is 0.07 µmol mol$^{-1}$. Calibration corrections calculated for $CH_4$ mole fractions have a mean of 2.8 ±3 nmol mol$^{-1}$ over the eleven sites and a yearly standard deviation of 0.7 nmol mol$^{-1}$ on average. Even if the corrections are quite homogeneous from instrument to instrument and over the course of a year, these values demonstrated the need for regular calibrations with standard references to

comply with WMO objectives of compatibility goals.

The data are corrected with the closest calibration equation in time existing before the data. As soon as there are calibration episodes before and after the considered data, the correction is made with a linear interpolation of the enclosing calibration equations. It is important to note that NRT data provided after 24 h will be automatically modified after a few weeks once the next calibration is available to estimate the temporal drift of the analyzer. If

no calibration equation is available within a period of 180 days to correct the data, the data are flagged as incorrect, and the explanation is added to the internal cumulative flag.

We have analyzed, for eleven monitoring stations, the differences in the $CO_2$ and $CH_4$ mole fractions processed in near-real time with the same dataset after calibration drift correction and manual validation by the PI. A posteriori verification of the NRT dataset is important to qualify this specific product, which is increasingly

requested by users. Understanding the reasons for differences between NRT and validated datasets will also help improve the automatic processing of the measurements. Figure 10 shows the differences for the hourly means. The most evident feature of the differences for all sites is the linear drift correction between two calibration sequences (≈ 2 to 4 weeks). In most cases (95%), the differences are within ±0.06 µmol mol$^{-1}$ for $CO_2$ and ±0.75 nmol mol$^{-1}$ for $CH_4$. The statistics of the validated minus NRT mole fractions are shown for each site in Figure

11. It is worth noting that for most of the stations, the median differences are less than or equal to zero. Only three instruments show a positive median difference for $CO_2$ (Lamto station—identified by the 3-letter code LTO with instrument #192, PDM #222, Ivittuut station—identified by the 3-letter code IVI with instrument #93) and one for $CH_4$ (IVI—#93). This means that almost all instruments have a tendency to drift positively; consequently, when a NRT dataset is revised after a few days or weeks with the new calibration sequence, its

value is slightly decreased. This tendency for a positive drift for $CH_4$ measurements by CRDS analyzers was also noticed by Yver-Kowk et al. (2015).



In addition to the data corrections due to instrumental drift, we also detect in Figure 10 some isolated events that present a different profile of variability, and there are also a few outliers. For example, not shown on this figure is a five-day period (10–15 July) at Mace Head station (#41) with very high differences between NRT and validated mole fractions: up to -25 $\mu$mol mol$^{-1}$ for $CO_2$ and -250 nmol mol$^{-1}$ for $CH_4$. This event corresponds to

the installation of a new calibration scale at Mace Head station, with erroneous values of the standard gases entered into the database. Consequently, the mole fractions calculated in NRT were wrong, and a few days were required to identify the problem and reprocess the dataset. Another example is the relatively constant differences observed at Finokalia station (identified by the 3-letter code FKL) from 5 to 20 June 2014: +0.09 $\mu$mol mol$^{-1}$ and +1.4 nmol mol$^{-1}$ for $CO_2$ and $CH_4$, respectively (Figure 10). This event corresponds to an error in the first

calibration performed at the installation of the station. The calibration episode was later rejected, and the subsequent calibration was therefore the only one used to correct the raw values, as explained previously. This issue may be difficult to detect immediately upon the start of a monitoring site because we lack references for evaluation. We also observe for some periods a relatively high random variability of the mole fraction differences for the Trainou station instrument (#108). This is due to the leakage of one valve that is used to

evacuate the liquid water from a water trap setup inside a refrigerator. This problem caused contamination for a few minutes. These contaminated values were used in the NRT data processing, whereas they were excluded after the quality control of the measurements performed by the station PI, which explains the differences between the two datasets.

### 5    Data time aggregation and associated metadata

Further processing consists of aggregating the data in time. The minute, hourly and daily means are computed for in situ data. The minute means and injection means are computed for tank data (calibration and target gases). As recommended by the World Data Centre for Greenhouse Gases (WDCGG, WMO, 2012), we calculate the means using data from the nearest time aggregation level and not always using the raw data. This implies that raw data are used to calculate minute averages, which are then used to calculate hourly averages and so on. For

each single averaged data point, we provide the number of data used to compute the average and the standard deviation. The measurement time associated with an average dataset corresponds to the beginning of the averaging period (e.g., the hourly means at 13:00 are calculated from the minute means from 13:00 to 13:59), which is also in line with the recommendation of WDCGG (WMO, 2012). The times provided to the users are always universal time. The time difference between local time and universal time is provided in the metadata of

the station.

Different data output formats can be provided to fit user needs. The files provided to the users always include the following information for each average mole fraction in dry air: time/date of the measurement, site and instrument identifiers, number of data and standard deviation, user flag and an internal identifier tracing all processing parameters. In addition, the header of the file provides metadata including the station coordinates, the

measurement calibration scale, the name of a contact person and the institute in charge of the monitoring program. More information (raw data, internal flags, etc.) is available upon request to the ATC data center.





## 6    Conclusion and perspectives

The provision of atmospheric greenhouse gas mole fractions in NRT is useful for early detection of anomalies, whether they are instrumental or geophysical, and data assimilation schemes. As part of the construction of the ICOS ATC data center, we have developed a framework for fast delivery (24 h) of the atmospheric greenhouse

gases dataset. The setup of the hardware and software needed for data collection, data processing, configuration of measurements and quality control of the time series have been performed over the past years in close collaboration with experimentalists in charge of running stations during the demonstrator phase of ICOS. The NRT processing chain was built on the expertise gained during previous European projects including CARBOEUROPE, Infrastructure for Measurements of the European Carbon Cycle (IMECC) and Global Earth

Observation and MONitoring (GEOMON). In the last few years, we moved from a situation in which each European station was performing its own data processing to the ICOS configuration with a central database and a set of software processing the raw data transferred from all ICOS sites daily. This configuration ensures better inter-comparison of the data. By analyzing data from eleven sites, we determined that the average calibration corrections applied in the data process by the ATC equals 1.7 ±0.3 μmol mol⁻¹ for $CO_2$ and 2.8 ±3 nmol mol⁻¹ for

$CH_4$. These biases are important to correct to avoid artificial gradients between stations that could lead to error in flux estimates when using atmospheric inversion techniques. Masarie et al. (2011), showed that a 1 μmol mol⁻¹ bias at a measurement tower in Wisconsin induced a response in terms of fluxes of 68 TgC/yr when using the carbon tracker inversion system (Peters et al., 2007). This flux represents approximately 10% of the estimated North American annual terrestrial uptake.

We have also evaluated that the average drift between two calibrations amounts to ±0.05 μmol mol⁻¹ and ±0.7 nmol mol⁻¹ for $CO_2$ and $CH_4$, respectively. Outliers may occur, which are generally associated with an error in the metadata information provided by the station PI (e.g., error in the attributed value of the calibration gas).

ICOS aims to maintain very high-precision measurements with a high level of data recovery, traceability, and fast delivery. Rapid access to processed data and their associated metadata, as well as a catalogue of data

products updated daily, is intended to facilitate the verification of the measurements. In 2013, 17.8 GB of data files and data products were viewed by users on the ICOS ATC website (https://icos-atc.lsce.ipsl.fr), which corresponds to more than 17,000 hits and more than 380,000 pages viewed. Traceability of the downloads, long-term archival and data policies beyond the scope of this paper are being designed in collaboration with the carbon portal of ICOS.

Thus far, the NRT dataset has been provided to the participants of the ICOS Preparatory Phase and the following projects: InGOS (http://ingos-atm.lsce.ipsl.fr/), ICOS-INWIRE (http://www.icos-inwire.lsce.ipsl.fr/) and MACC-III/COPERNICUS (http://www.copernicus-atmosphere.eu/d/summary/macc/gac/verif/ghg/icos/). The format of the files provided to the users was adapted to their needs, and the identifier which allows for the traceability of the measurements is part of the compulsory information. The MACC-III project is using the $CO_2$ data in NRT

time to evaluate their assimilation and forecasting system developed at the European Centre for Medium-range Weather Forecasts (Agusti-Panareda et al., 2014). In another study, the authors performed a $CH_4$ inversion to test



the ability of the European network of atmospheric observations to detect the leakage of an offshore oil platform at Elgin Field, North Sea (Berchet et al., 2013).

The continuous enhancement of automatic processing is important, and new developments are in progress. This includes the evaluation of spike detection algorithms that would allow the automatic identification of data being

significantly influenced by local processes. Another perspective is to interface the database with the electronic logbooks of the station operations (maintenance, troubleshooting, etc.), as a support of the quality control of the time series. One important issue is the estimation of time-varying uncertainties based on regular measurements of the target gases, comparison of in situ and flask measurements, and analysis of specific tests. Evaluation of algorithms to estimate random and systematic errors is performed by the INGOS and ICOS-INWIRE European

projects, and we have begun to transfer some of them in the ICOS data processing.

**Acknowledgements**

The authors acknowledge the scientists and engineers from the stations contributing to the development of the ICOS data center. Their regular feedback and comments are essentials for the success of the project. We wish to

acknowledge the valuable comments of M. Steinbacher, H. Chen, L. Rivier, J.D. Paris, M. Delmotte and N. Schneider and the entire staff of the ICOS Atmospheric Data Center. This study was funded in part by the European Commission under the EU Seventh Research Framework Programme through ICOS (grant agreement n°211574), and ICOS-INWIRE (grant agreement n°313169).





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



| Valid data | Invalid data | Definition | Data level involved |
|:---:|:---:|---|---|
| **U** | **N** | Automatic quality control | Raw, minutely, hourly |
| **O** | **K** | Manual quality control | Raw, minutely, hourly |
| **R** | **H** | Backward propagation of manual quality control from hourly data to minutely and raw data | Raw, minutely |

**Table 1:** List of user flags. The user flag is instrument independent. Behind the validity status of the data, each set of flags conveys additional meaning. Automatic quality control flags imply that no expert has manually inspected the data yet, whereas manual quality control flags imply that an expert has manually inspected the data. The backward propagation of manual quality control flags, imply that an expert has performed manual inspection of the corresponding aggregate data but not the data directly.





| Flag | Description | Status of the data |
|---|---|---|
| S | Station not working properly | Invalid |
| I | Instrument not working properly | Invalid |
| d | Air distribution system not working properly | Invalid |
| T | Tank issue | Invalid |
| F | Stabilization/flush period | Invalid |
| L | Inlet leakage | Invalid |
| E | External disturbance near the station | Invalid |
| C | Calibration issue | Invalid |
| A | Maintenance with contamination | Invalid |
| X | Instrument out of order | Invalid |
| G | Data out of range | Invalid |
| | | |
| Q | QA operation | Valid |
| M | Maintenance | Valid |
| Z | Non-background conditions | Valid |

**Table 2:** List of descriptive flags. The descriptive flag is instrument independent and is picked from a limited given list. The flag is case sensitive. Multiple flags (i.e., letters) can be set simultaneously on a single value.

5   There is a list to be used for invalid data and one to be used for valid data.



| Sequence definition | | |
| --- | --- | --- |
| Definition of the calibration sequence | Example: MHD #41 | |
|   Valve port connections of calibration tanks | 4 tanks on ports 3, 4, 5 and 6 | |
|   Tank measurement duration | 20 min | |
|   Number of calibration cycles | 4 | → 320 min |
| Definition of the ambient/target sequence | | |
|   Valve port connections of sampling line(s) and target tanks | Port 1: in situ, 2: short term target | |
|   Duration of ambient air measurement | 660 min | |
|   Duration of short-term target measurement | 20 min | |
|   Duration of long-term target measurement | 20 min | |
|   Duration of reference measurement | 0 min | |
|   Long-term target measured before the ambient/tgt sequence | yes | |
|   Long-term target measured after the ambient/tgt sequence | no | |
|   Short-term target measured before the ambient/tgt sequence | no | |
|   Short-term target measured after the ambient/tgt sequence | no | |
|   Number of ambient/target cycles | 62 | → 43,420 min |
| Definition of the intercomparison sequence | | |
|   Valve port connections of sampling line(s) and tanks | - | |
|   Duration of intercomparison tank measurement | 20 min | |
|   Duration of short-term target measurement | 0 min | |
|   Duration of reference measurement | 0 min | |
|   Duration of ambient air measurement | 0 min | |
| Definition of the overall sequence | 1 × Ambient/ target seq. | |
| | 1 × Calibration seq. | |

**Table 3:** Definition of a measurement sequence. As an example, we show the configuration for the instrument installed at Mace Head station (identified by the 3-letter code MHD), Ireland.



| Instrument parameterization configuration | |
|---|---|
| Stabilization duration | Example: MHD #41 |
|   In situ gas | 5 min |
|   Target gas | 15 min |
|   Calibration gas | 15 min |
|   Reference gas | 0 min |
| Physical parameters | |
|   Cavity pressure | 139.8–140.2 torr |
|   Cavity temperature | 44.98–45.02 °C |
|   Outlet valve opening | 15,000–55,000 |
| Processing parameters | |
|   In situ gas: interval filtering | 350–500 µmol mol$^{-1}$ ($CO_2$) <br> 1700–2500 nmol mol$^{-1}$ ($CH_4$) |
|   Target gas: humidity filtering | < 0.05 µmol mol$^{-1}$ ($CO_2$) <br> < 0.2 nmol mol$^{-1}$ ($CH_4$) |
|   Calibration gas: humidity filtering | < 0.05 µmol mol$^{-1}$ ($CO_2$) <br> < 0.2 nmol mol$^{-1}$ ($CH_4$) |
| Correction parameters | |
|   In situ gas: humidity correction | a=-0.00982 b=-2.393e$^{-4}$ |
|   In situ gas: calibration correction | cf. calibration parameters below |
|   Target gas: humidity correction | a=-0.00982 b=-2.393e$^{-4}$ |
|   Target gas: calibration correction | cf. calibration parameters below |
|   Calibration gas: humidity correction | a=-0.00982 b=-2.393e$^{-4}$ |
| Calibration computing parameters | |
|   Standard deviation for minute means of calibration gas measurement | < 0.08 µmol mol$^{-1}$ ($CO_2$) <br> < 0.8 nmol mol$^{-1}$ ($CH_4$) |
|   Standard deviation for cycle means of calibration gas measurement | < 0.06 µmol mol$^{-1}$ ($CO_2$) <br> < 0.5 nmol mol$^{-1}$ ($CH_4$) |
|   Minimum number of cycles per tank | 2 |
|   Minimum tanks to compute the fitting equation | 3 |
|   Number of cycles for the stabilization period | 1 |
|   Fitting equation degree | 1 |

**Table 4:** List of the parameters used for the automatic processing of $CO_2$ and $CH_4$ mole fractions by CRDS analyzers. The humidity filtering applied to the tank measurements consists of checking the absolute difference between the wet value and the computed dry value against the defined threshold. The parameters are specific to the instrument considered.





| Internal flag name | Criteria | Example (#41, MHD) |
|---|---|---|
| Stabilization | Data acquired during the stabilization period | Ambient air: 5 min<br>Target gases: 15 min<br>Calibration gases: 15 min |
| Cavity pressure | Cavity pressure not in the valid interval | 139.8–140.2 torr |
| Cavity temperature | Cavity temperature not in the valid interval | 44.98–45.02 °C |
| Outlet | Outlet not in the valid interval | 15,000–55,000 |
| Humidity | The difference due to the humidity between raw data and corrected data is above the threshold | $CO_2$: 0.05 µmol mol$^{-1}$<br>$CH_4$: 0.20 nmol mol$^{-1}$ |
| Filter | Data not in the valid interval | $CO_2$: 350–500 µmol mol$^{-1}$<br>$CH_4$: 1700–2500 nmol mol$^{-1}$ |
| Calibration | No valid calibration | - |
| Unitary data | No unitary data available | - |
| Minute standard deviation | Standard deviation for calibration minute data above the threshold | $CO_2$: 0.08 µmol mol$^{-1}$<br>$CH_4$: 0.80 nmol mol$^{-1}$ |
| Cycle standard deviation | Standard deviation for calibration injection data above the threshold | $CO_2$: 0.06 µmol mol$^{-1}$<br>$CH_4$: 0.50 nmol mol$^{-1}$ |
| MaxDeltaDurationTank | The time interval between 2 successive calibration tanks is too large | 1 min |
| NbTank | The number of tanks for the calibration is below the minimum required | 3 tanks |
| TankMinDuration | The measurement duration for a tank is below the configured minimum | 10% of the defined duration for the given type of tank (target or calibration) |
| TankMaxDuration | The measurement duration for a tank is above the configured maximum | 10% of the defined duration for the given type of tank (target or calibration). The calibration is not rejected, but a warning email is sent to notify the PI that more gas than expected is used up. |
| NbCycle | The number of cycles for a tank measurement during calibration is below the minimum required | 2 cycles |
| SequenceCompleteness | The calibration sequence is incomplete | See calibration sequence definition |
| Quality control | Manual rejection flag set up by the station PI | - |
| Backwards quality control | Propagation of a manual flag set up by the station PI on an aggregated value (e.g., the hourly mean) to all data used for averaging (e.g., the minute and raw data) | - |

**Table 5:** List of internal flags for instrument type CRDS Picarro model G2301. The example provided in the third column corresponds to the configuration of instrument #41 at Mace Head station set up on 14 May 2009.





| CH₄ raw data from 2014-01-01 to 2014-12-11 | | | | | |
|---|---|---|---|---|---|
| Site | ID | User flag | Internal flag | N data | % |
| MHD | 41 | O | Stabilization, Quality control, Backwards quality control | 23 | 0.00 |
| MHD | 41 | N | Stabilization, Cavity temperature | 11 | 0.00 |
| MHD | 41 | N | Stabilization, Cavity pressure | 657 | 0.01 |
| MHD | 41 | N | Stabilization | 38833 | 0.75 |
| MHD | 41 | N | Cavity temperature | 268 | 0.01 |
| MHD | 41 | N | Cavity pressure, Outlet | 1 | 0.00 |
| MHD | 41 | N | Cavity pressure | 201 | 0.00 |
| Site | ID | User flag | Internal flag | | |
| MHD | 54 | K | Stabilization, Quality control | 1520 | 0.02 |
| MHD | 54 | N | Stabilization, Cavity temperature | 995 | 0.01 |
| MHD | 54 | K | Stabilization, Cavity pressure, Quality control | 25 | 0.00 |
| MHD | 54 | N | Stabilization, Cavity pressure, Cavity temperature | 4 | 0.00 |
| MHD | 54 | N | Stabilization, Cavity pressure | 586 | 0.01 |
| MHD | 54 | N | Stabilization | 143243 | 1.51 |
| MHD | 54 | K | Filter, Quality control, Backwards quality control | 6 | 0.00 |
| MHD | 54 | K | Filter, Quality control | 5421 | 0.06 |
| MHD | 54 | N | Filter | 18704 | 0.20 |
| MHD | 54 | N | Cavity temperature | 5583 | 0.06 |
| MHD | 54 | K | Cavity pressure, Quality control, Backwards quality control | 10 | 0.00 |
| MHD | 54 | K | Cavity pressure, Quality control | 322 | 0.00 |
| MHD | 54 | N | Cavity pressure, Cavity temperature | 1 | 0.00 |
| MHD | 54 | N | Cavity pressure | 753 | 0.01 |
| Site | ID | User flag | Internal flag | | |
| OPE | 91 | N | Stabilization, Cavity temperature | 71 | 0.00 |
| OPE | 91 | N | Stabilization, Cavity pressure, Outlet | 282 | 0.01 |
| OPE | 91 | N | Stabilization, Cavity pressure | 40377 | 1.58 |
| OPE | 91 | N | Stabilization | 409067 | 15.97 |
| OPE | 91 | N | Filter | 80 | 0.00 |
| OPE | 91 | N | Cavity temperature | 54 | 0.00 |
| OPE | 91 | K | Cavity pressure, Quality control | 5 | 0.00 |
| OPE | 91 | N | Cavity pressure, Outlet | 6085 | 0.24 |
| OPE | 91 | N | Cavity pressure | 145 | 0.01 |

**Table 6:** Examples of user and internal flags that were attributed to raw data from CRDS Picarro instruments in 2014. The two last columns provide the number of raw data that have been attributed an internal flag or combination of internal flags and the corresponding percentage in the dataset. Most of the data have no internal flag, indicating that there is no anomaly detected.





| |
|---|
| Site;Site id;Street;City;Zip code;Country;Latitude;Longitude;Altitude;Time zone;Owners;Owner codes;Contact emails;Sampling heights;Sampling ids |
| Observatoire pérenne de l'environnement;OPE;Route de l'observatoire; Houdelaincourt; 55130;France;48.5619;5.5036;390;+1;Andra–Agence nationale pour la gestion des déchets radioactifs;; marc.delmotte@lsce.ipsl.fr,olivier.laurent@lsce.ipsl.fr,michel.ramonet@lsce.ipsl.fr,sebastien.conil@andra.fr;10.0 ,50.0,120.0;1,2,3 |
| |
| Site;Year;Month;Day;Hour;Minute;Second;DecimalDate;co2;Stdev;NB;Flag;InstId;SamplingHeight;Quality;Aut oDescriptiveFlag;ManualDescriptiveFlag |
| OPE;2015;04;06;00;00;00;2015.26027397;413.813;0.611;16;U;91;10;44402;I,F-1;<br>OPE;2015;04;06;01;00;00;2015.26038813;413.922;0.290;16;U;91;10;44402;I,F-1;<br>OPE;2015;04;06;02;00;00;2015.26050228;415.767;0.562;16;U;91;10;44402;I,F-1;<br>OPE;2015;04;06;03;00;00;2015.26061644;416.919;0.481;16;U;91;10;44402;I,F-1;<br>OPE;2015;04;06;04;00;00;2015.26073059;417.937;0.426;16;U;91;10;44402;I,F-1;<br>OPE;2015;04;06;05;00;00;2015.26084475;420.584;0.571;15;U;91;10;44402;;<br>OPE;2015;04;06;06;00;00;2015.26095890;419.648;0.906;16;U;91;10;44402;I,F-1;<br>OPE;2015;04;06;07;00;00;2015.26107306;415.025;0.787;16;U;91;10;44402;I,F-1;<br>OPE;2015;04;06;08;00;00;2015.26118721;410.413;0.914;16;U;91;10;44402;I,F-1;<br>… |

**Table 7:** Example of a file provided to the MACC-2 project in near-real time (24 h). The first block represents the metadata of the station, and the second block contains hourly means of $CO_2$ mole fractions in dry air for one day (Observatoire Pérenne de l'Environnement station, identified by the 3-letter code OPE, instrument #91).





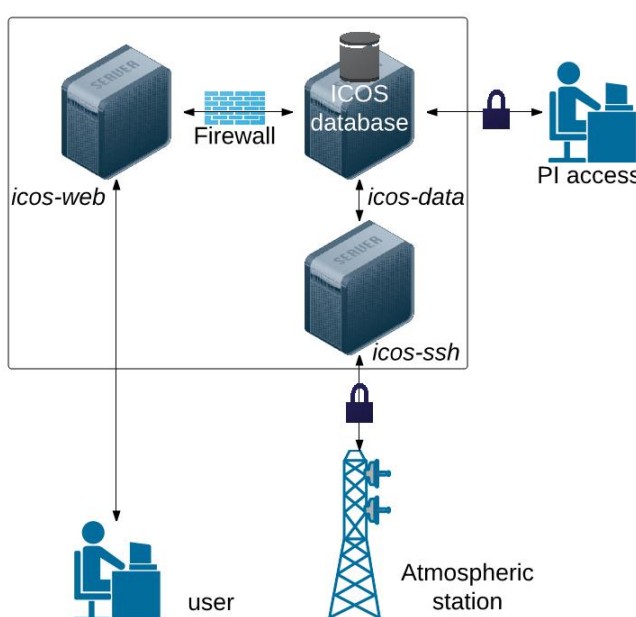

**Figure 1.** Schematic view of ICOS ATC network infrastructure.





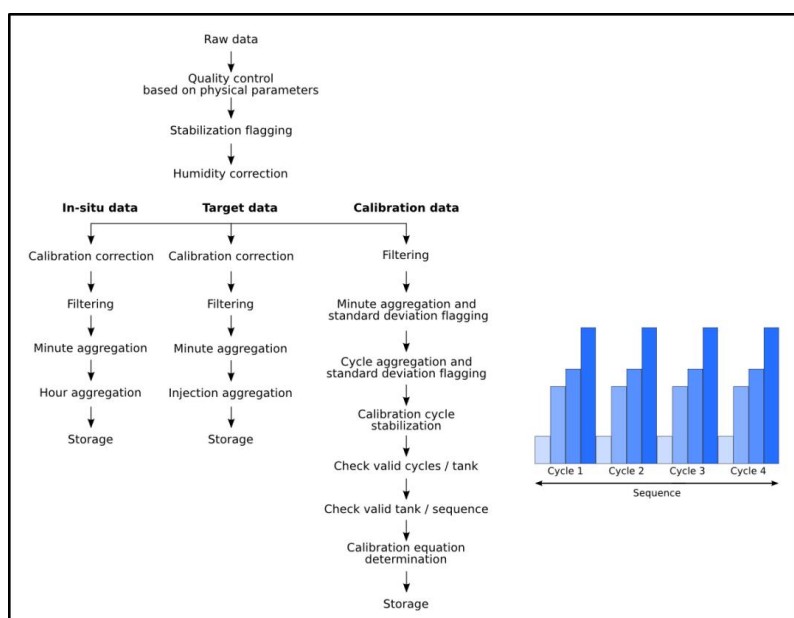

**Figure 2.** Automatic data processing of $CO_2$ and $CH_4$ data at ICOS ATC. We consider three types of data: 'in situ' corresponding to ambient air, 'target' when a cylinder filled with a reference gas is measured, and 'calibration' when calibration cylinders are measured.



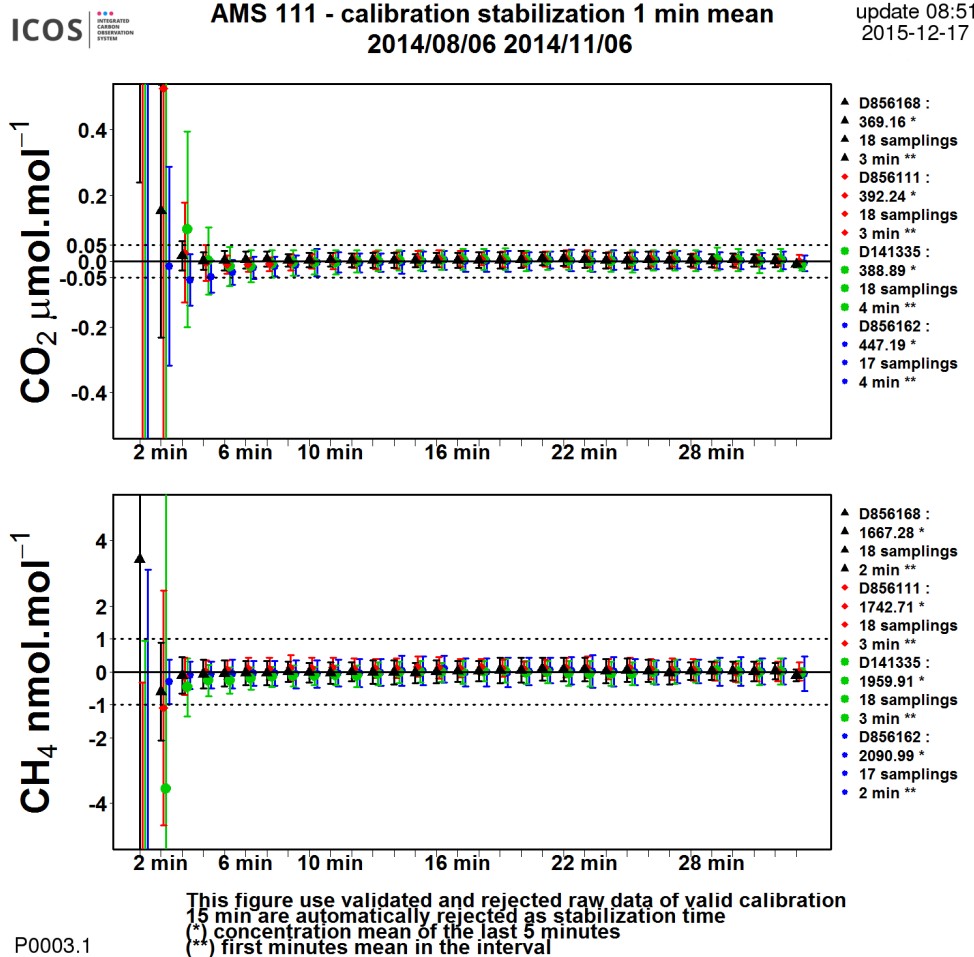

**Figure 3.** $CO_2$ (above) and $CH_4$ (below) mole fraction differences between each minute and the last minute of the target gas measurement period (30 min in this case) at the Amsterdam Island station (identified by the 3-letter

5   code AMS). The differences are averaged for all target gas measurements from 6 Aug. to 6 Nov. 2014. The number of injections or samplings during this period is provided for each of the four target gases on the right. The minutes provided on the right of the graph for each gas correspond to the minute when the difference decreases below the horizontal dashed lines chosen as half the WMO-recommended compatibility for Northern hemisphere sites ($\pm 0.05$ $\mu$mol mol$^{-1}$ for $CO_2$ and $\pm 1$ nmol mol$^{-1}$ for $CH_4$).





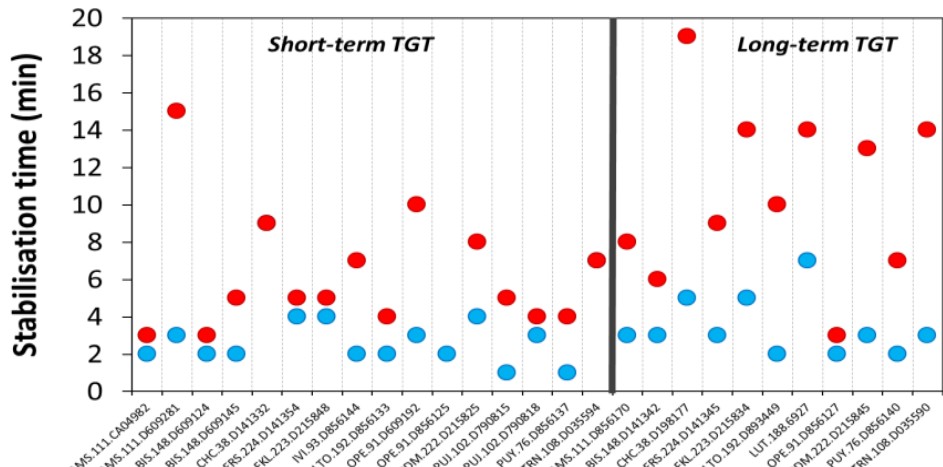

**Figure 4.** Average stabilization times (in minutes) estimated to have a difference from the last minute of the target gas measurement of less than ±0.05 μmol mol$^{-1}$ for $CO_2$ (in red) and ±1 nmol mol$^{-1}$ for $CH_4$ (in blue). The time is calculated for several instruments indicated on the x-axis, and the left side of the figure shows short-term target gas measurements, whereas the right side shows the long-term target measurements, which are less frequent.





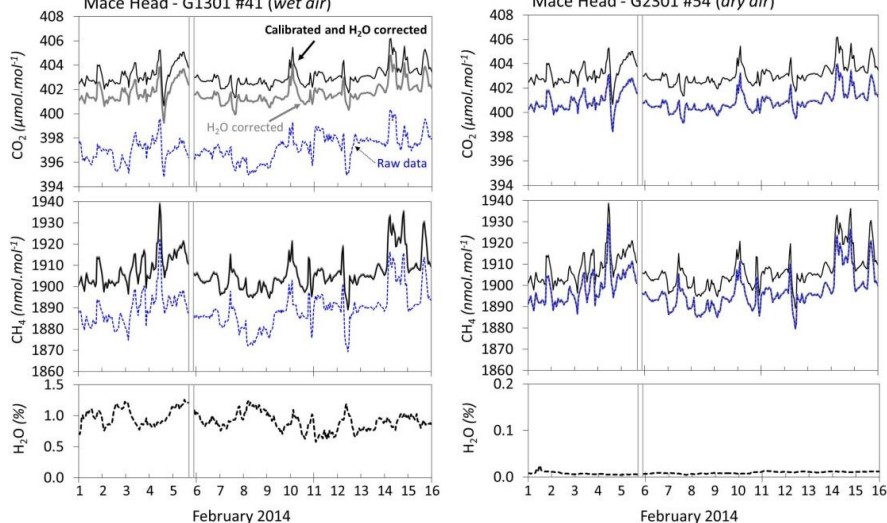

**Figure 5.** $CO_2$ (above), $CH_4$ (middle) and $H_2O$ (below) mole fractions observed at Mace Head in February 2014 with two CRDS analyzers. Left: analyzer Picarro model G1301 (#41) measuring wet air. Right: analyzer Picarro model G2301 (#54) measuring dry air. For $CO_2$ and $CH_4$ plots, the blue dashed lines correspond to the raw data, the gray lines correspond to the raw data corrected for water vapour, and the thick black line corresponds to the calibrated mole fractions in dry air.





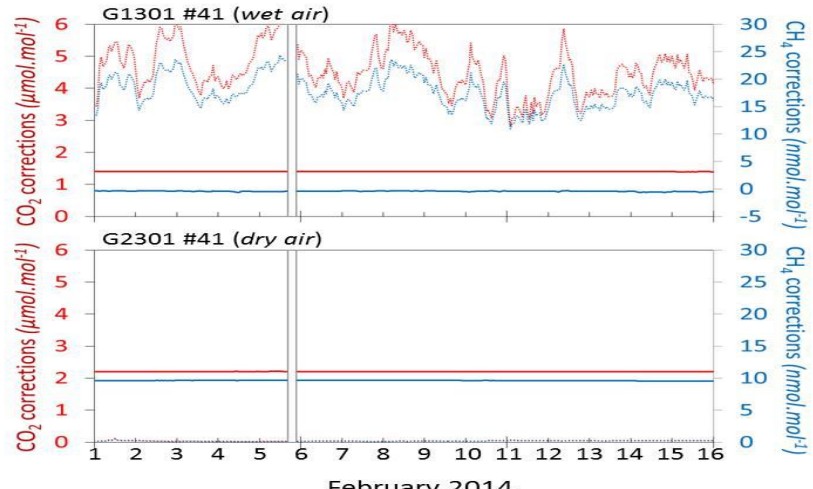

**Figure 6.** Water vapour corrections (dashed lines) and WMO calibration corrections (thick lines) applied to $CO_2$ (red) and $CH_4$ (blue) mole fractions for two CRDS analyzers used at Mace Head station (above: #41 measuring wet air; below: #54 measuring dry air).

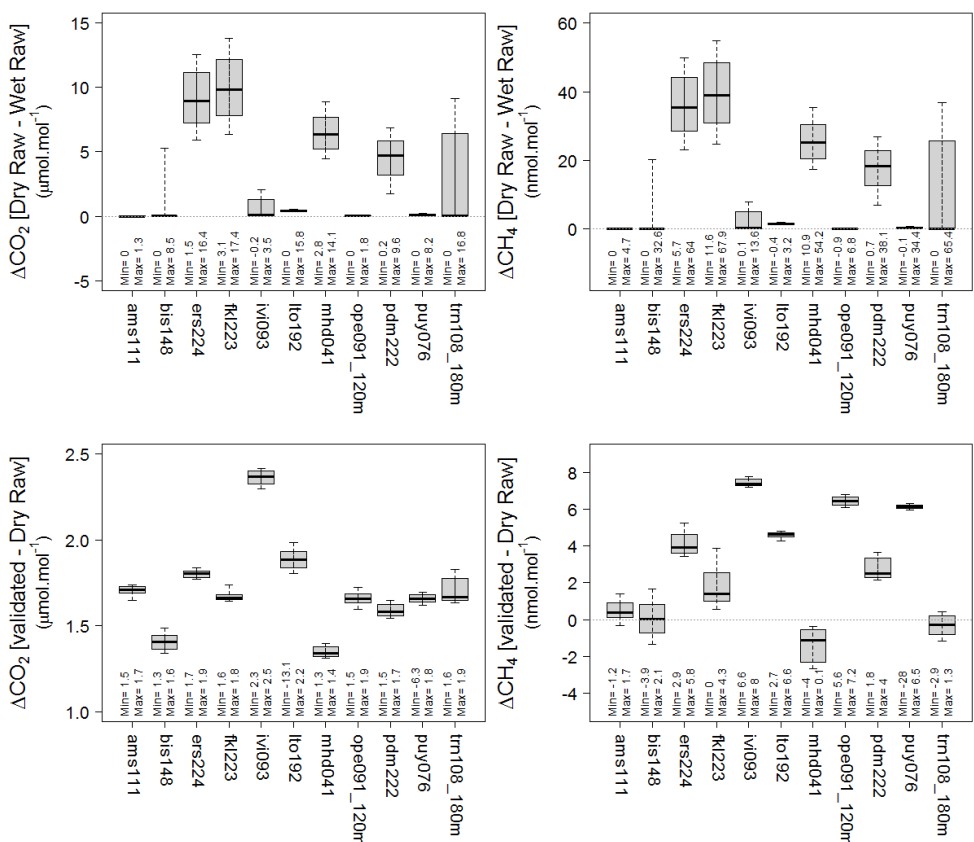

5    **Figure 7.** Synthesis of the water vapour (above) and calibration (below) corrections applied to eleven
instruments in 2014 for hourly mean $CO_2$ (left) and $CH_4$ (right) mole fractions. The length of the box represents
the interquartile range, the horizontal line represents the median, and the low and high whiskers show 10% and
90% percentiles, respectively. Numbers below the boxplots give the maximum and minimum corrections. It
should be noted that the calibration corrections depend on the calibration settings of the analyzers.





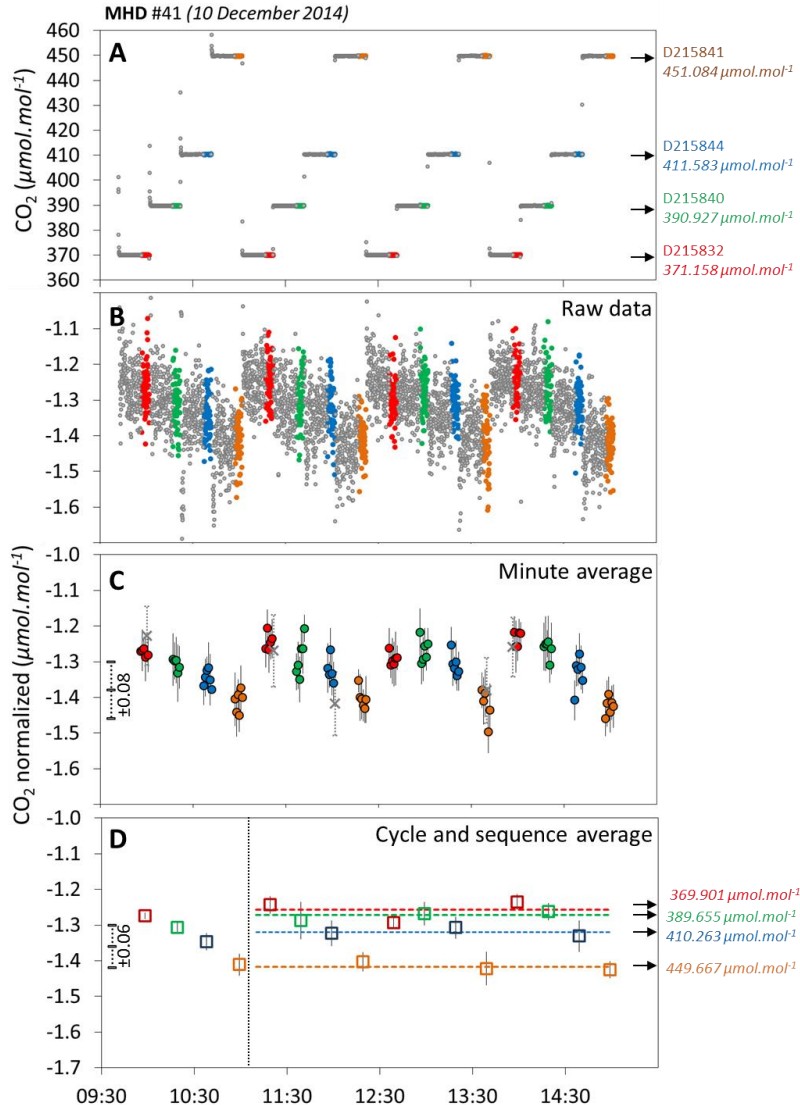

**Figure 8.** Details of a $CO_2$ calibration performed at Mace Head station (instrument #41) on December 10, 2014. A. Raw $CO_2$ data measured for 4 calibration tanks analyzed 4 times in 20 min. Gray points show the rejected values during the stabilization period (i.e., flushing period). Values indicated on the right give the tank ID and their attributed mole fractions on a WMOx2007 scale. B. Same as A for $CO_2$ mole fraction differences between measured values and attributed WMO values. C. Same as B for minute averages. Gray crosses show rejected values due to a standard deviation higher that the threshold value (vertical bar on the left). D. Cycle (squares) and calibration sequence averages (dashed lines and values on the right). The first cycle is rejected as a stabilization period.




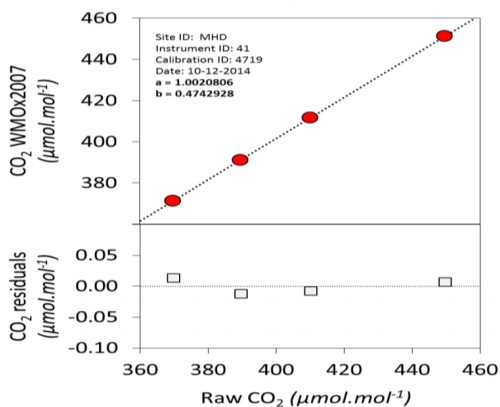

**Figure 9.** Linear fit of the $CO_2$ calibration detailed in Figure 8. Coefficients a and b of the fit are shown in bold

5    characters. The lower plot shows $CO_2$ residuals from the linear regression.





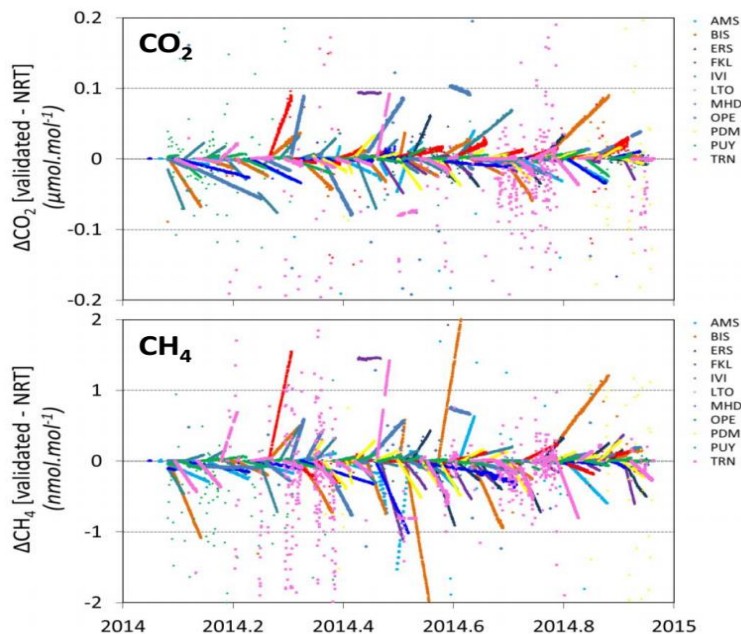

5   **Figure 10.** $CO_2$ (above) and $CH_4$ (below) mole fraction differences between the validated and the near-real time

values at eleven stations in 2014. Most of the differences correspond to the drift between two calibrations, which

cannot be considered in real time. Each point corresponds to an hourly average.





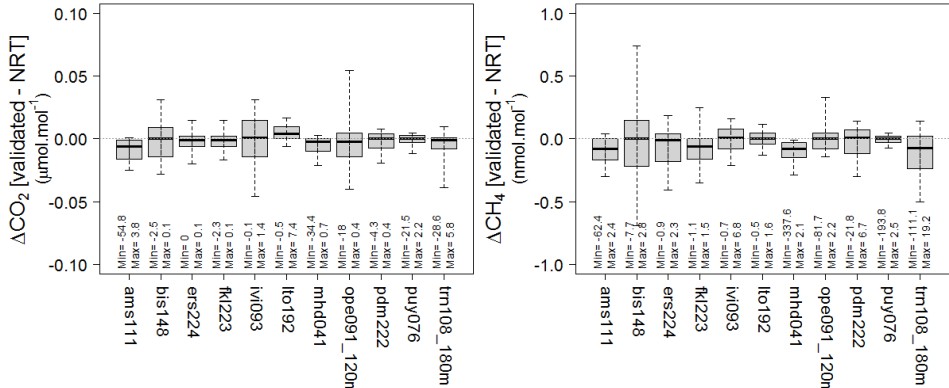

**Figure 11.** Statistics of the validated minus NRT differences of hourly means $CO_2$ (left) and $CH_4$ (right) mole
fractions. Each of the eleven box-and-whisker plots describes the differences for monitoring stations in 2014.
The length of the box represents the interquartile range, the horizontal line represents the median, and the low
and high whiskers show the first and ninth deciles, respectively. The numbers below the boxplots give the
maximum and minimum differences.