# Peer review of "Automatic processing of atmospheric CO2 and CH4 mole fractions at the ICOS Atmospheric Thematic Center"

_Atmospheric Measurement Techniques, 2016_

## Referee Comment (RC1) · J. Klausen (Referee) · 20 May 2016

1. Does the paper address relevant scientific questions within the scope of AMT? - The monitoring of greenhouse gases is a fundamental requirement for sound scientific understanding of the drivers of climate change, and equally relevant for mitigation and adaption.

2. Does the paper present novel concepts, ideas, tools, or data? - The European research infrastructure ICOS is a distributed facility with centralized data management. The paper describes in detail how the data management and quality assurance procedures are set up. Findings are illustrated by illustrative examples. The treatment of uncertainties along the traceability chain should be more rigorous.

[Figure]

3. Are substantial conclusions reached? - The network-wide application of the same procedures permits the early detection of data quality issues. The main conclusion is that the system works as intended, facilitating quick remedy in case of problems.

4. Are the scientific methods and assumptions valid and clearly outlined? - The treatment of uncertainties along the traceability chain should be more rigorous. The method of simple linear regression (e.g., Figure 9) does not consider the uncertainty of the standards against which is calibrated. To do this properly, a fully weighted regression technique, as described by Press et al, Numerical Recipes, Chapter 15.3 should be considered - Further, the calibration cycle suggested (cf. Figure 8A) could be improved by randomization, such that the sequence within the cycle changes and possible correlations are more easily detected.

5. Are the results sufficient to support the interpretations and conclusions? - Yes

6. Is the description of experiments and calculations sufficiently complete and precise to allow their reproduction by fellow scientists (traceability of results)? - Yes

7. Do the authors give proper credit to related work and clearly indicate their own new/original contribution? - Yes

8. Does the title clearly reflect the contents of the paper? - Yes

9. Does the abstract provide a concise and complete summary? - Yes

10. Is the overall presentation well structured and clear? - Yes

11. Is the language fluent and precise? - Yes, a few systematic linguistic errors would be caught by a native Speaker

12. Are mathematical formulae, symbols, abbreviations, and units correctly defined and used? - Yes, except for the unit Mo/day and Go/day (p3L14, p3L20, and throughout manuscript)

13. Should any parts of the paper (text, formulae, figures, tables) be clarified, reduced,

combined, or eliminated? - p11L16: A reasoning should be given to explain why this approach is considered superior (or why it even makes a difference) - p12L7: This statement is important and should already be used in the introduction/motivation. - Figure 7: The text on p8 should distill the main message better. This reviewer reads from the text and figure mainly that water-vapor-corrected data exhibit less of a bias than the raw (wet) data. Is this the message? - Figure 10: The Figure is extremely busy, and this reviewer finds it hard to identify the example discussed in the text. It is suggested to perhaps present most of the points in gray (losing their identity) but instead highlight those that are being discussed in color. Moreover, interesting features like

(blue dots), where the drift is towards smaller bias , should be discussed.

14. Are the number and quality of references appropriate? - Yes

15. Is the amount and quality of supplementary material appropriate? - No supplementary material is provided, but all information can be obtained upon authentification from the processing centre.

Jörg Klausen/2016-05-20

Please also note the supplement to this comment:
http://www.atmos-meas-tech-discuss.net/amt-2016-53/amt-2016-53-RC1-supplement.pdf

---

## Referee Comment (RC2) · Anonymous Referee #2 · 31 May 2016

The paper does represent a substantial contribution to scientific progress in terms of new measurement techniques. It describes (in detail) the data processing procedures of the ICOS ATC, and this description is required as more countries join the European ICOS network. It is of vital importance to ensure the comparability and consistency of data submitted to an integrated EU network.

The paper is well written and the scientific methods applied are appropriate. The figures and tables are in the main well presented and add to the understanding of the results explained in the text.

Specific comments: P4 L18. The problem with keeping a target tank for 10-20 years is that the mole fractions will be so far removed from ambient levels (assuming the current

growth rate for CO2 and CH4), that instrument non-linearity effects might dominate the comparison.

P8 L9-19. To what extent can the differences be attributed to wet-dry sampling as opposed to instrumental differences between a G1301 and a G2301?

P8 L20-29. The text and Figure 7 are confusing as they show a comparison of data before and after water corrections – however, some of the instruments use physical dryers (which could bias the data), however, the information detailing which sites are using dryers is not given.

General comments: P3 L1-2. Text does not read very well, try – "Because this paper is focused on CO2 and CH4, only analysers deployed in the monitoring network that measure these gases have been considered". P3 L4. Don't Los Gatos off-axis instruments meet ICOS requirements? P3 L28. Text does not read very well, try "uses an open-source content management system framework (Drupal)". P4 L2. Replace are with have been P4 L4. Replace are with have also been P4 L27. Each instrument does not flag their raw data, the instrument operators flag the data, or setup the parameters for automatic flagging or someone at the ICOS ATC sets this up? P5 L 7 ICOS-MSA, 2014 is not listed in the references. P6 L 5. Change "we are scanning" to "each data point is scanned for" P6 L8. Change the "we" so something else. P7 L3. Shouldn't the unique identifier be #111 and also in Figure 3, it should be AMS #111 no AMS 111. P7 L6. It looks more like stabilization is reached after 4-6 mins in the AMS #111 example. Figure 4 also indicates that ∼20% of CO2 values are not reached within a 10 minute period. P8 L23. States that the Mace Head instrument is close to zero because it is using a dryer system , however the MHD #41 instrument shown in Figure 7 does not use a dryer? P9 L1. WMO scale for CH4 was updated in 2015 to WMO X2004A P9 L13. What happens if the values plotted do not follow a liner function? Or the calibration sequence mole fraction range do not cover the ambient mole fraction range? P15 L19. What does this refer to in the Manning reference? K., S., R., S., L. P., S., J., T., Y., T., R. L., V., 20 A., V., F., and Worth, D P15 L 21. Should the reference link be:

http://cucumbers.uea.ac.uk/documents/2014_InGOS_NA3_Cucumbers_Report.pdf
P16 L 6. Yver Kwok, C

---

## Author Comment (AC1) · 20 Jul 2016

4 - The treatment of uncertainties along the traceability chain should be more rigorous. The method of simple linear regression (e.g., Figure 9) does not consider the uncertainty of the standards against which is calibrated. To do this properly, a fully weighted regression technique, as described by Press et al, Numerical Recipes, Chapter 15.3 should be considered

We agree with the comment that the calculation of uncertainties must be developed. As part of the ICOS project we have organized working groups to handle specific issues. There are currently two working groups focusing respectively on the calibration strategy, and the water vapor corrections. For both issues we want to come up with

a more rigorous approach for uncertainties calculations, which will be implemented in the data processing. Regarding the calibration, the group will assess the uncertainties associated to the fitting curve (in which weighted technique should be implemented), time interpolation of the calibration, and possible non linearity. A comment about those further developments of the data processing has been added in the conclusion P13 L27: Within the ICOS project research actions are ongoing for a better assessment of the calibration strategy and the water vapor correction, and their associated uncertainties. The outputs of these studies will be implemented later in the data processing to improve the current data corrections and uncertainties estimates.

4 - Further, the calibration cycle suggested (cf. Figure 8A) could be improved by randomization, such that the sequence within the cycle changes and possible correlations are more easily detected.

For most cases the calibration are done with calibrations gases analyzed in rising order, even though this is not mandatory for ICOS, and it is the responsibility of the station PI to configure the order of the cylinders. One test has been done showing that the order of measurement of the 6 cylinders had not significant effect on the results (see additional figure 1).

12. Are mathematical formulae, symbols, abbreviations, and units correctly defined and used? - Yes, except for the unit Mo/day and Go/day (p3L14, p3L20, and throughout manuscript).

The units have been checked and MB/day and GB/day are used throughout manuscript.

13.Should any parts of the paper (text, formulae, figures, tables) be clarified, reduced, combined, or eliminated?

- p11 L16: A reasoning should be given to explain why this approach is considered superior (or why it even makes a difference)

[Figure]

Not being sure which approach the referee is pointing out, we assume it concerns the double quality control, both automatic and manual. It is very hard to completely automatize the quality control, there are always some specific cases which can only be seen by expert's eyes who may have additional information at hand; therefore the need of a double control. The PI has to provide codified reasons for invalidating data or useful information for validating data (the list of such reasons, called 'descriptive flag', can be found in Table 2).

The following text has been added P11 L33:

This example shows the importance of the expert examination; it is very hard to completely automatize the quality control and the PI may have additional information at hand to help define the status of the data. However, when invalidating data the PI has to provide codified reasons (the list of such reasons, called 'descriptive flag', can be found in Table 2).

- p12 L7: This statement is important and should already be used in the introduction/motivation

We have moved the statement to the introduction P2 L12:

The NRT processing chain was built on the expertise gained during previous European projects including CARBOEUROPE, Infrastructure for Measurements of the European Carbon Cycle (IMECC) and Global Earth Observation and MONitoring (GEOMON). NRT is defined here as on a daily basis.

- Figure 7: The text on p8 should distill the main message better. This reviewer reads from the text and figure mainly that water-vapor-corrected data exhibit less of a bias than the raw (wet) data. Is this the message?

The text and the figure deal with the level of the water vapor correction (difference between the data with and without the $H_2O$ correction) depending on the water vapor level . For both wet and dry air the water vapor correction is applied. The amplitude

of the correction is a function of the H2O concentration. In dry air, the level of the water correction is close to zero (it depends on the level of residual water vapor). In term of uncertainties, the water vapor correction may introduce a correction bias for wet air depending to the water vapor level (imperfection in the determination of the water vapor correction function). On the other hand the method used to dry the air (e.g. Nafion membrane) may also introduce a bias. We are currently preparing another paper to assess those uncertainties associated with the water vapor.

We have clarified the text as follows on P8 L23:

The water vapor corrections shown in Figure 7 correspond to the difference between data with and without the H2O correction (amount of water vapor correction), and not to a measurement or correction bias. These corrections are needed to convert humid air mole fractions in dry air mole fractions. However, any error in the water vapor correction would introduce a bias in the resulting dry air mole fractions, whose amplitude would depend on the H2O concentration. The determination of a specific correction for each instrument by the ATC will minimize the bias associated with humid air measurements. Conversely, drying the air (e.g. using a Nafion membrane) may also cause a measurement bias by contamination of the sampled air. The evaluation of these biases is underway at the ATC and will be published separately.

- Figure 10: The Figure is extremely busy, and this reviewer finds it hard to identify the example discussed in the text. It is suggested to perhaps present most of the points in gray (losing their identity) but instead highlight those that are being discussed in color. Moreover, interesting features like (blue dots), where the drift is towards smaller bias, should be discussed.

For a better view of the figure, we have added to the Figure 10 a one month zoom with only 3 sites (Finokalia - FKL, Lamto – LTO and Puy de Dôme – PUY)). Few events were already discussed for Mace Head, Finokalia and Trainou, and we have added a short description of the zoom portion of the figure. The legend has also been enlarged.

The figure 10 caption has been updated as follows:

Figure 10. $CO_2$ (above) and $CH_4$ (below) mole fraction differences between the validated and the near-real time values at eleven stations in 2014 (left), and at three stations (Finokalia - FKL, Lamto – LTO and Puy de Dôme – PUY) in June 2014 (right). Most of the differences correspond to the drift between two calibrations, which cannot be considered in real time. Each point corresponds to an hourly average.

We have added the following precisions to the manuscript:

P11 L2: At AMS station we see a reverse slope for a short period (2 weeks) on early July 2014, with drift going towards smaller bias over time. This is due to a revision of the calibration performed on July 1st, after the correction of an erroneous injection of one calibration gas.

P11 L25: The zoom over June 2014 (Figure 10 right) also shows small wavelet in the $CO_2$ differences at Lamto station. This feature is related to the strong diurnal cycle observed at this tropical site (typically 50 ppm). The correction applied to the data being concentration dependent the difference between NRT and validated dataset also display the diurnal cycle.

[Figure]

[Figure]

[Figure]

**Fig. 1.** Additional figure 1

**Fig. 2.** Updated manuscript figure 10

---

## Author Comment (AC2) · 20 Jul 2016

- P4 L18. The problem with keeping a target tank for 10-20 years is that the mole fractions will be so far removed from ambient levels (assuming the current growth rate for CO2 and CH4), that instrument non-linearity effects might dominate the comparison.

The idea is to install long term targets (LTT) with relatively high concentrations. Currently the recommendations for the LTT concentrations are ranging from 450 to 470 ppm for CO2 and 2100 to 2200 ppb for CH4. Considering current mean concentrations of about 400 ppm and 1900 ppb for CO2 and CH4 respectively, and the associated trends of about +2.5 ppm/yr and +10 ppb/yr this would lead to mean concentrations of about 450 ppm and 2100 ppb in 20 years from now. So the LTT cylinders will still

be in the range of atmospheric concentrations. Regarding the calibration, in order to maintain the suite of 3-4 reference gases in line with the atmospheric range over time, the plan is to change one cylinder every 5 years, rather than changing the full suite after 20 years.

- P8 L9-19. To what extent can the differences be attributed to wet-dry sampling as opposed to instrumental differences between a G1301 and a G2301?

We have one reference gas measured regularly (twice a day) by the two instruments to assess their agreement when measuring dry air. We attribute the difference not explained by this reference gas (dry) to the water vapor correction.

- P8 L20-29. The text and Figure 7 are confusing as they show a comparison of data before and after water corrections – however, some of the instruments use physical dryers (which could bias the data), however, the information detailing which sites are using dryers is not given.

We have added the following precisions to the manuscript P8:

The instruments operated at Amsterdam Island, Biscarrosse, Lamto, the Observatoire Pérenne de l'Environnement and Puy de Dôme were measuring dry air, whereas the Trainou instrument was successively operated in the two configurations (wet and dry) in 2014.

- P3 L1-2. Text does not read very well, try – "Because this paper is focused on CO2 and CH4, only analysers deployed in the monitoring network that measure these gases have been considered".

We have corrected the sentence P3:

Because the paper is focused only on CO2 and CH4, only analyzers deployed in the monitoring network that measure these gases have been considered.

- P3 L4. Don't Los Gatos off-axis instruments meet ICOS requirements?

[Figure]

So far only the CRDS/Picarro analyzers have been labeled for CO2 and CH4 measurements in ICOS. The LGR analyzer has been labeled as well for CO measurements. The ICOS specifications can be found in the following document: https://icos-atc.lsce.ipsl.fr/?q=filebrowser/download/8681

We have evaluated on CO2/CH4 analyzer commercialized by LGR (model GGA 24 EP ) but the repeatability of the measurements were not satisfactory (tested on 2 different units), as well as the feedback from the manufacturer to try solving this issue.

- P3 L28. Text does not read very well, try "uses an open-source content management system framework (Drupal)".

We have corrected the sentence P3 L30:

This server hosts the ATC website and uses an open-source content management system framework (Drupal).

- P4 L2. Replace are with have been

We have corrected the sentence P4:

Specific processing chains have been developed for each type of trace gas analyzer, but the general framework remains the same.

- P4 L4. Replace are with have also been

We have corrected the sentence P4 L7:

Similar chains have also been developed for measurements of other ICOS parameters such as meteorological variables or radon but are not described in detail in this article.

- P4 L27. Each instrument does not flag their raw data, the instrument operators flag the data, or setup the parameters for automatic flagging or someone at the ICOS ATC sets this up?

The sentence has been rewritten P4 L29:

Because ways and conditions to automatically validate raw data may differ from an instrument model to another, the list of internal flags are instrument dependent.

- P5 L 7 ICOS-MSA, 2014 is not listed in the references.

The reference has been added on P5 L10:

ICOS Atmospheric Station Specifications, ICOS, 2015

- P6 L 5. Change "we are scanning" to "each data point is scanned for"

We have corrected the sentence P6 L7:

In the case of the CO2/CH4 analyzers currently used in the ICOS network, each raw data point is scanned for three parameters: the cavity pressure, the cavity temperature and the outlet valve opening.

- P6 L8. Change the "we" so something else.

We have corrected the sentence P6 L10:

Consequently, for each single data point, the values of the parameters are checked against a valid interval or threshold.

- P7 L3. Shouldn't the unique identifier be #111 and also in Figure 3, it should be AMS #111 no AMS 111.

We have corrected the definition P7 L4:

identified by the 3-letter code AMS, instrument #111

- P7 L6. It looks more like stabilization is reached after 4-6 mins in the AMS #111 example. Figure 4 also indicates that 20% of CO2 values are not reached within a 10 minute period.

As indicated in the manuscript for the long term target, which are analyzed less frequently, the stabilization time is clearly longer.

We have corrected the numbers according to your comments P7 L8:

When looking at measurements of short-term and long-term target gases from several sites (Figure 4), one can see that stabilization is very often reached within 4-6 min, but more time may be needed for the equilibration of the long-term target.

- P8 L23. States that the Mace Head instrument is close to zero because it is using a dryer system, however the MHD #41 instrument shown in Figure 7 does not use a dryer?

This is correct. We are running two instruments in MHD, one measuring dry air (#54) and the other one measuring wet air (#41). However this normal configuration was switched during few weeks to evaluate the water vapor corrections. The reference of the instrument measuring dry air (#54) has been added.

P8 L31: Several instruments are operated with a drier system, and the water vapor corrections are consequently close to zero, as shown for the Mace Head station (for the instrument #54).

- P9 L1. WMO scale for CH4 was updated in 2015 to WMO X2004A

The change of reference scale for CH4 is underway. For the ICOS network this action is organized by the calibration center (CAL).

- P9 L13. What happens if the values plotted do not follow a liner function? Or the calibration sequence mole fraction range do not cover the ambient mole fraction range?

The calibration scales are chosen to cover the atmospheric range. In addition all the analyzers are evaluated at the ICOS/ATC laboratory with an extended calibration scale (e.g. 300 to 500 ppm for CO2), prior to their installation in the field.

However, for the events with atmospheric concentrations exceeding the highest reference gas, e.g. in case of pollution event, we may have to increase the uncertainties. This estimation of the uncertainties is still under evaluation by a dedicated ICOS worknone

ing group. The residuals from the linear fit will be used for the estimation of uncertainties. The additional figure 2 shows the residuals from a linear fit performed during the initial test period of instruments at ATC/ICOS laboratory (using the same set of tanks). During the automatic processing, both linear and second order functions are computed. By default the linear fit is used to process the ambient air measurements, but the differences between the two functions may be used in the estimation of the uncertainties.

- P15 L19. What does this refer to in the Manning reference? K., S., R., S., L. P., S., J., T.,Y., T., R. L., V., 20 A., V., F., and Worth, D

The reference has been corrected P16 L16:

Manning, A. C., Jordan, A., Levin, I., Schmidt, M., Neubert, R. E. M., Etchells, A., Steinberg, B., Ciais, P., Aalto, T., Apadula, F., Brand, W. A., Delmotte, M., Giorgio di Sarra, A., Hall, B., Haszpra, L., Huang, L., Kitzis, D., van der Laan, S., Langenfelds, R. L., Leuenberger, M., Lindroth, A., Machida, T., Meinhardt, F., Moncrieff, J., Morgu ′ Äś, J. A., Necki, J., Patecki, M., Popa, E., Ries, L., Rozanski, K., Santaguida, R., Steele, L.P., Strom, J., Tohjima, Y., Thompson, R.L., Vermeulen, A., Vogel, F., and Worth, D.: Final report on CarboEurope "Cucumber" intercomparison programme. Available at http://cucumbers.uea.ac.uk/documents/2014_InGOS_NA3_Cucumbers_Report.pdf, 2009.

- P15 L 21. Should the reference link be: http://cucumbers.uea.ac.uk/documents/2014_InGOS_NA3_Cucumbers_Report.pdf

The reference link has been corrected. P16 L16:

http://cucumbers.uea.ac.uk/documents/2014_InGOS_NA3_Cucumbers_Report.pdf, 2009.

- P16 L 6. Yver Kwok,

The reference has been updated P17 L6:

Yver Kwok, C., Laurent, O., Guemri, A., Philippon, C., Wastine, B., Rella, C. W., Vuillemin, C., Truong, F., Delmotte, M., Kazan, V., Darding, M., Lebègue, B., Kaiser, C., Xueref-Rémy, I., and Ramonet, M.: Comprehensive laboratory and field testing of cavity ring-down spectroscopy analyzers measuring $H_2O$, $CO_2$, $CH_4$ and CO, Atmos. Meas. Tech., 8, 3867–3892, 2015.
* * *
[Figure]

[Figure]

[Figure]

[Figure]

**Fig. 1.** Additional figure 2